# Phase-Selective Microwave Assisted Synthesis of Iron(III) Aminoterephthalate MOFs

**DOI:** 10.3390/ma13061469

**Published:** 2020-03-23

**Authors:** Ana Arenas-Vivo, David Avila, Patricia Horcajada

**Affiliations:** 1Advanced Porous Materials Unit, IMDEA Energy. Av. Ramón de la Sagra 3, 28935 Móstoles-Madrid, Spain; ana.arenas@imdea.org; 2Department of Inorganic Chemistry, Chemical Sciences Faculty, Complutense University of Madrid, 28040 Madrid, Spain; davilabr@ucm.es

**Keywords:** microwave synthesis, metal-organic frameworks, porous solids, iron, aminoterphthalate, phase selection

## Abstract

Iron(III) aminoterephthalate Metal-Organic Frameworks (Fe-BDC-NH_2_ MOFs) have been demonstrated to show potential for relevant industrial and societal applications (i.e., catalysis, drug delivery, gas sorption). Nevertheless, further analysis is required in order to achieve their commercial production. In this work, a systematic synthetic strategy has been followed, carrying out microwave (MW) assisted hydro/solvothermal reactions to rapidly evaluate the influence of different reaction parameters (e.g., time, temperature, concentration, reaction media) on the formation of the benchmarked MIL-101-NH_2_, MIL-88B-NH_2_, MIL-53-NH_2_ and MIL-68-NH_2_ solids. Characterization of the obtained solids by powder X-ray diffraction, dynamic light scattering and transmission electron microscopy allowed us to identify trends to the contribution of the evaluated parameters, such as the relevance of the concentration of precursors and the impact of the reaction medium on phase crystallization. Furthermore, we presented here for the first time the MW assisted synthesis of MIL-53-NH_2_ in water. In addition, pure MIL-101-NH_2_ was also produced in water while MIL-88-NH_2_ was the predominant phase obtained in ethanol. Pure phases were produced with high space-time yields, unveiling the potential of MW synthesis for MOF industrialization.

## 1. Introduction

Metal-Organic Frameworks (MOFs) are an interesting and recent family of hybrid crystalline solids based on different inorganic units connected with polydentate ligands [1]. Among other properties (e.g., optical, magnetic, electrochemical), they are notable for their impressive porosity [2,3] and their modulable chemical and topological structure [4,5]. As a result, they have an open door for their use in plenty of striking applications, both at industrial and social level, such as fluid storage and separation [6,7], catalysis [8,9], and drug delivery [10,11], among others [12].

Among the great diversity of polydentate ligands, the linear terephthatate derivatives (BDC; including the 1,4-benzene dicarboxylic acid or H_2_BDC and its amine substituted form H_2_BDC-NH_2_) are very popular, leading to the formation of a large variety of highly porous MOF structures based on diverse cations (e.g., Cu, Zn, Al, Cr, Sc) [13]. For instance, based on the earth abundant, non-toxic and redox active iron, the series of polymorphic iron(III) terephthalates MIL-53 [14], MIL-68 [15], MIL-88B [16] and MIL-101 [17] (MIL stands for Matériaux Institut Lavoisier) has been described and widely explored [18]. Briefly, MIL-101 or [Fe_3_O(OH)(BDC)_3_(H_2_O)_2_] is based on iron(III) octahedra trimers (Fe_3_-µ_3_-oxo clusters) and the BDC, creating a rigid three-dimensional (3D) cubic MTN-zeotype structure with exceptional mesoporosity (Brunauer, Emmett & Teller surface (S_BET_) ~ 2500 m^2^·g^−1^, pore volume (V_p_) ~ 2.4 cm^3^·g^−1^) with two mesocages of diameters of 29 and 34 Å, accessible through pentagonal and hexagonal windows of 12 and 15 × 16 Å, respectively) [17]. A polymorph, also based on iron(III) octahedra trimers, is the flexible microporous MIL-88B structure [Fe_3_O(OH)(BDC)_3_(H_2_O)_2_] [16] exhibiting a 3D hexagonal network with interconnected cages and pore channel system able to reversibly modify its size (up to 9 Å; with a unit cell volume variation up to 120%) as a function of different external stimuli (e.g., adsorbate, pressure, temperature). Two additional crystalline phases can exist based on the BDC ligand but different inorganic unit, here iron(III) octahedra chains FeO_4_(OH)_2_: MIL-53 and MIL-68, both with the chemical formula [Fe(OH)(BDC)]. MIL-53 is a flexible 3D network with a flexible diamond shaped 1D channels [14]. This structure reaches a variation of a 40% in volume in between open (up to 8.5 Å) and closed form, depending on the guest molecules, pressure and temperature. Finally, MIL-68 is a polymorph of MIL-53, having a rigid orthorhombic structure with hexagonal (~15 Å) and triangular (~5 Å) shaped pores (S_BET_ ~ 670 m^2^·g^−1^) and its amino substituted version has not been reported for iron(III) so far [15]. These benchmarked solids have proven exceptional sorption, catalytic and photocatalytic properties [18].

Consequently, due to the interest of this family of MOFs, subsequent research on Fe-BDC-NH_2_ synthesis is fundamental not only for understanding the underlying formation mechanisms but also to promote facile synthetic protocols scalable for commercialization and MOF industrial application [19]. With this is mind, microwave synthesis (MW) of MOFs has been proposed as an alternative to conventional hydro- or solvothermal reactions due to several advantages: (i) energy efficiency, (ii) fast crystallization (increment in number of reaction sites), (iii) phase selectivity, (iv) high yields, (v) variety of morphologies, (vi) particle size control, (vii) lower temperatures and reaction times [20]. The conjunction of this assets dramatically increases production due to the homogeneous energy input, compared to a traditional batch reactor, which can even be enhanced by reaction stirring [21]. In addition, industrial production in the order of T·year^-1^ can be achieved under MW-assisted continuous flow synthesis [22]. In consonance, this technology is efficiently exploited nowadays in the production of several organic chemicals such as drugs or polymers [23,24].

Even at its infancy, MW method has become a resourceful tool for the preparation of MOFs and for their activation/purification (removing pore filling species) [20,25]. Fe-based MOFs have not been an exception and there are previous reports that focus on the MW synthesis of a particular Fe-MOF phase (mostly without substitution of the H_2_BDC ligand) [17,26,27,28,29,30]. However, as modest variations of the method lead to vast different synthetic results, a systematic study on the different variables affecting MW synthesis of iron (III) aminoterephthalate phases is still lacking. In this work, a systematic synthetic strategy has been followed, carrying out MW-assisted hydro/solvothermal reactions to rapidly evaluate the influence of different reaction parameters (e.g., time, temperature, concentration, reaction media) on the formation of the benchmarked MIL-101-NH_2_, MIL-88B-NH_2_, MIL-53-NH_2_ and MIL-68-NH_2_ solids. Crystallinity analysis as well as phase identification has been carried out by means of powder X-ray diffraction (PXRD), while particle size was investigated by dynamic light scattering (DLS) measurements and transmission electron microscopy (TEM) observation. In addition, important parameters (i.e., reaction yield and space-time yield) have been determined in order to prove the usefulness of MW synthesis for the future high scale production of Fe-BDC-NH_2_ MOFs.

## 2. Materials and Methods

### 2.1. Synthesis

Fe-BDC-NH_2_ MOF structures (for codes see Appendix A) were synthesized from iron(III) chloride hexahydrate (FeCl_3_·6H_2_O, Sigma Aldrich, 97%, Madrid, Spain), 2-aminoterephthalic acid (H_2_BDC-NH_2_, Acros Organics, 99%, Geel, Belgium), ethanol (EtOH, Labkem, 96%, Barcelona, Spain), absolute ethanol (Labkem), *N,N*-dimethylformamide (DMF, Acros, 99+%, Riad, Saudi Arabia), hydrochloric acid (HCl, J.T. Baker, 37–38%, Loughboroug, United Kingdom) and deionized water.

Different synthetic conditions were investigated (see Appendix A) for the microwave (MW) assisted-hydro/solvothermal synthesis of Fe-BDC-NH_2_ MOFs. In general, a reaction mixture of 4 mL was loaded in a 10 mL glass vial (G10, Anton Paar GmbH, Graz Austria) and sonicated for 1 min in an ultrasound bath (Branson 1800, Emerson, St. Louis, Missouri, United States) prior to being sealed and placed in a Monowave-300 MW (maximum power 300 W, Anton Paar GmbH, Graz Austria). The reactor vial in the MW was heated to the reaction temperature as fast as possible with different power impulses, maintained for a predetermined time and cooled down to 65 °C by an air flow prior extraction of the vial from the MW. The mixture was stirred during the MW reaction at 600 rpm at all times. As synthesized samples were recovered by centrifugation (12045 g for 10 min) and washed three times with absolute EtOH (2 g·L^−1^ under stirring) and recovered by centrifugation (12045 g for 10 min; activated sample, *act.*). Samples were stored wet in absolute EtOH prior to further analysis.

### 2.2. Characterization

Crystal phase of all samples was verified using powder X-ray diffraction (PXRD). PXRD patterns were collected using a conventional PANalytical Empyrean powder diffractometer (PANalytical Lelyweg, Almelo, Netherlands, *θ*–2*θ*) using λ_Cu_ K_α1_, and K_α2_ radiation (λ = 1.54051 and 1.54433 Å). PXRD diagrams were carried out with a 2ϴ scan between 3–35° with a step size of 0.013° and a scanning speed of 0.1°·s^−1^.

Particle size was monitored via dynamic light scattering (DLS using a Zetasizer Nano series Nano-ZS (Malvern Instruments, Worcestershire, UK). The solids were dispersed in the liquid media (EtOH) at a concentration of 0.1 mg·mL^−1^ using an ultrasound tip (UP400S, Hilscher, Teltow, Germany) at 20% amplitude for 10 s.

Transmission electron microscopy (TEM) images were taken with a JEM 1400 (Jeol, Tokyo, Japan) with a 120 kV acceleration voltage (point resolution 0.38 nm). For TEM studies, 1 mg of sample was dispersed in 10 mL of absolute EtOH and sonicated with an ultrasound tip (UP400S, Hilscher, Teltow, Germany) at 20% amplitude for 10 s. For observation, 1 µL of the prepared solution was dropped over a copper TEM grid coated with holey carbon support film (Lacey Carbon, 300 mesh, copper, approx. grid hole size: 63 µm, TED PELLA Redding, CA, USA). Particle size was monitored via statistic counting (*n* = 70–100) with ImageJ (Version 1.8.0).

To address the MW assisted reaction performance two parameters have been defined: reaction yield (wt.%, Equation (1) and space-time yield (STY, kg·m^−3^·d^−1^, Equation (2))
(1)$$Yield\;\left(\%\right)=\frac{m_{\mathrm{Fe}-\mathrm{BDC}-{\mathrm{NH}}_{2}}}{m_{\mathrm{theoretical}}}\times 100$$
where m_FeBDC-NH2_ is the experimental mass of the Fe-BDC-NH_2_ MOF obtained (determined after drying at 100 °C) and m_theoretical_ is the calculated theoretical amount of the determined phase that could be obtained, according to the metal used (Appendix A), both in kg.
(2)$$STY=\frac{m_{\mathrm{Fe}-\mathrm{BDC}-{\mathrm{NH}}_{2}}}{V\times t}$$
where m_FeBDC-NH2_ is the experimental mass of the Fe-BDC-NH_2_ MOF obtained (kg), V is the reaction volume (m^3^) and t is the reaction time (day).

## 3. Results

H_2_O as reaction medium. The FeCl_3_/H_2_BDC-NH_2_ system in water was studied under two different temperatures (100 and 150 °C) varying the iron precursor concentration (0.02, 0.05, 0.1 and 0.2 M). As an example, some of these conditions have been resumed in Table 1 (for further information see Appendix A). Under the analyzed reaction conditions, two different phases have been formed: MIL-101-NH_2_ and MIL-53-NH_2_ (Figure 1, for PXRD see Appendix A). Note that it is typical to find here crystallized unreacted H_2_BDC-NH_2_ in the as synthesized (*a.s.*) samples due to the poor solubility of the ligand in water. However, the linker excess is removed during the following activation step (see experimental section). As expected, while lower temperature led to the formation of the mesoporous MIL-101-NH_2_ (kinetically favored phase), at higher temperature the thermodynamic flexible microporous phase MIL-53-NH_2_ was formed. In addition, there is a bisectrix on the T (°C)/[Fe] (M) crystallization diagram in which a mixture of phases is obtained, revealing this concentration dependency: in brief, higher concentrated reactions favor the formation of chains of corner-sharing FeO_4_(OH)_2_ octahedra leading to MIL-53-NH_2_, whereas diluted conditions promote the formation of the trimer Fe_3_-µ_3_-oxo clusters and, therefore, MIL-101-NH_2_ crystallization.

Furthermore, to evaluate the influence of the pH on the FeCl_3_/H_2_BDC-NH_2_ system, the addition of the strong acid HCl was investigated within the region of the bisectrix, leading to mixture of MIL-53-NH_2_ and MIL-101-NH_2_ phases (T = 100 °C and [Fe] = 0.05 M, pH variation from 1.5 in the reaction mixture to 1 with HCl). At first glance, HCl presence increased the crystallinity of the Fe-H_2_BDC-NH_2_ phases (see PXRD in Appendix A). Furthermore, one can qualitatively observe by PXRD that MIL-101-NH_2_ proportion seems to decrease (2θ = 4–6°) while MIL-53-NH_2_ content seems to increase (2θ = 8.8 and 10.1°) with the addition of HCl, until only MIL-53-NH_2_ is produced with higher HCl contents (200 µL).

On the whole, in water, increasing reaction temperature or increasing acidity shifts the equilibrium FeCl_3_/H_2_BDC-NH_2_ system to the more favorable thermodynamic phase: MIL-53-NH_2_.

EtOH as reaction medium. MIL-88B-NH_2_ was the predominant phase observed of the FeCl_3_/H_2_BDC-NH_2_ system when using ethanol as reaction medium (see Figure 2 and PXRD in Appendix A). All the reaction parameters with EtOH as solvent are resumed in Appendix A. Although, at lower temperature (100 °C) and longer times (≥20 min), MIL-88B-NH_2_ was transformed to a poorly crystalline MIL-101-NH_2_. In an attempt to obtain the thermodynamic MIL-53-NH_2_ phase as well in ethanol, the reaction temperature was increased to 180 °C. However, except for very short times (5 min), the increment of the reaction temperature only led to the formation of α-Fe_2_O_3_ (rhombohedral hematite as identified by PXRD; Appendix A). This indicates that Fe_3_-µ_3_-oxo clusters are less thermodynamically favored than the iron(III) oxide under high temperatures. In addition, particle size experienced insignificant variation regardless of the conditions (Appendix A), remaining below 300 nm with low polydispersity indexes (PdI < 0.3), according to DLS measurements.

DMF as reaction medium. The different reaction parameters using DMF can be consulted in SI Appendix A. In contrast with water and EtOH, the study of the FeCl_3_/H_2_BDC-NH_2_ system in DMF in MW-assisted solvothermal synthesis did not lead to the formation of any solid at T = 100 °C. Even at 150 °C, short times under MW (5 min) only led to the formation of poorly crystalline solids. Increasing the time promotes the formation of the MIL-101-NH_2_, favored phase in DMF as it was present at all tested concentrations in DMF (see Figure 3 and PXRD Appendix A). This need of extra-temperature could be explained because of the higher activation energy for the Fe_3_-µ_3_-oxo cluster to bond H^-^BDC-NH_2_ in DMF, compared to previously studied solvents (i.e., water and ethanol). Remarkably, at a concentration range 0.05–0.1 M, MIL-101-NH_2_ coexists with its polymorph MIL-88B-NH_2_, while at higher concentration (0.2 M) chain-based MIL-53-NH_2_ appears. Results suggest that DMF is a more complex reaction medium, being the only one in which three Fe-BDC-NH_2_ MOF phases were obtained.

## 4. Discussion

In the course of this study, 46 individual reaction mixtures (with replicates to ensure reproducibility) were carried out in three different protic and aprotic polar solvents (H_2_O, EtOH and DMF) using the MW-assisted solvothermal method. Water and ethanol have been selected as good examples of green chemistry reaction solvents. DMF has been selected as a typical reaction medium traditionally used in MOF solvothermal synthesis. Remarkably, the MW irradiation is a simple procedure that enables the fast and efficient attainment of three different Fe-BDC-NH_2_ crystalline phases by slight modifications on the synthetic conditions, reaching high reaction yields and space-time-yield (STY) comparable to those of industrial process in the market in very short times. Thus, the influence of different parameters (i.e., solvent, temperature, reaction time, molar ratio of metallic precursor and organic ligand, concentration of the reaction mixture and the addition of HCl) on the product formation is here discussed as a function of the sample purity and crystallinity, crystals dimension (including size distribution), reaction yield and STY. The purity and crystallinity of the MW-assisted experiments are discussed on the basis of PXRD (see Supporting information (SI) Section 2) and are schematically represented above in crystallization diagrams. The proportion of the starting reagents and the conditions of synthesis are given in the Appendix A.

H_2_O as reaction medium. Using water, pure MIL-101-NH_2_ is only obtained at low temperature and concentration (100 °C, 0.02 M), while single MIL-53-NH_2_ crystallizes at higher concentrations and temperature (0.1 M; and at 0.2M with both 100 and 150 °C). In addition, the formation of pure MIL-101-NH_2_ in water gave to an extremely high reaction yield (almost complete reaction ~100 %) with however, lower STY associated with the diluted starting concentration (3850 kg·m^3^·d^−1^ at 0.02 M). MIL-53-NH_2_, besides being the thermodynamically stable phase, was produced as a pure phase in lower yields (<70%; see Appendix A). However, considering the higher concentration of the reaction mixture, STY values reached 10400 kg·m^3^·d^−1^, being competitive with the reported production of different MOFs (STY for Basolites^©^ ~3000 kg·m^3^·d^−1^) [19,31].

Particle size determination, by means of DLS, gave a rough estimation of the particle size of 300 ± 100 nm, with no clear dependence with temperature neither with concentration. Here, it should be taken into account that DLS provides information of hydrodynamic size, which can lead to misinterpretation in the case of non-spherical particles, such as MIL-53-NH_2_, and with a mixture of different phases, as in this study. In addition, the pure MIL-53-NH_2_ phase obtained in water with the highest STY (10400 kg·m^3^·d^−1^; [Fe] = 0.2 M, T = 100 °C) was analyzed by TEM (see Appendix A). There can be seen well-defined submicrometric rhombohedral MIL-53-NH_2_ particles. The resulting average particle size was of 330 ± 50 nm (*n* = 70) with additionally a monodisperse distribution (PdI = 0.11), in accordance with previously observed in DLS (260 ± 70 nm Appendix A). To the best of our knowledge, this is the first report of a MW-assisted synthesis of pure submicronic iron(III) aminoterephtalate MIL-53-NH_2_ up to date [29,32].

Considering the tests carried out in HCl ([Fe] = 0.05 M, T = 100 °C), the results unveiled the beneficial effect of the strong acid both (Appendix A): (i) on the crystallization of MIL-53-NH_2_; higher HCl amounts (≥100 µL) favored the formation of MIL-53-NH_2_ over MIL-101-NH_2_; and (ii) on modulating and promoting the growth of a better crystallized MIL-101-NH_2_ (at lower acidic concentrations; 50 µL). This may be related to the better control of the kinetics rate reaction, associated to the deprotonation equilibrium of the carboxylate ligand (H_2_BDC-NH_2_ ⇌ H^-^BDC-NH_2_ + H^+^ ⇌ BDC-NH_2_ + H^+^). For a better analysis of the HCl presence, the water system FeCl_3_/H_2_BDC-NH_2_ with and without acid at 100 °C and [Fe] = 0.05 M was further analyzed by TEM (Appendix A). Micrographs revealed that in the absence of HCl, small rounded and undefined octahedra along the quaternary axis (80 ± 10 nm, *n* =150), corresponding to the MIL-101-NH_2_, were the predominant phase (Appendix A). It is important to outstand the small particle size obtained. A previous report using lower concentrations [Fe] = 0.02, and longer times attained the phase with ~300 nm (seen in TEM) [17]. Upon the addition of 100 µL of acid, although the octahedra of MIL-101-NH_2_ seem better defined and well-faceted, its amount significantly decreased, increasing in the meantime the content of larger MIL-53-NH_2_ flakes (190 ± 25 nm). Due to the mixture of phases and the absence of spherical particles, DLS measurements determined a high polydispersity (PdI > 0.3) and bigger particle size than the one determined by TEM. Note here that the reaction in presence of HCl was almost complete, with yields of around 100% (see Appendix A), but the increment in reaction volumes produces a decrease of the STY (from 8500 to 6500 kg·m^3^·d^−1^, when in the presence of 200 µL of acid). Despite this, a mixture of phases continued to appear up to high HCl volumes (200 µL), which limits the accurate estimation of the proportion of each of them.

EtOH as reaction medium. As previously stated, MIL-88B-NH_2_ is the predominant phase in EtOH and might be an interesting candidate for industrial production (see Figure 4). Nevertheless, it is important to notice that increasing MOF precursors concentration did not directly imply higher yields. In the case of reactions at 150 °C, the highest reaction yields were reached at lower concentration (100 and 85% at 0.02 and 0.05 M, respectively), being the more efficient reactions. However, considering the final production, despite the lower yields obtained with higher concentrated solutions (35% at 0.2 M), the maximum STY was performed at higher concentration (6200 kg·m^3^·d^−1^). Therefore, there is a compromise solution to maximize reaction yield and the final product amount, that according to results is achieved at T = 100 °C, [Fe] = 0.1 M and 5 min of reaction. (see reaction yields and STY in Appendix A).

Due to its higher STY (16000 kg·m^-3^·d^−1^) and crystallinity (see PXRD Appendix A), the sample synthetized at T = 100 °C and [Fe] =0.1 M was selected for further analysis with TEM. TEM micrographs (Appendix A) exhibited the characteristic bipyramidal hexagonal prims of MIL-88B-NH_2_, confirming the phase determined previously by PXRD. The needles length (l) as determined by TEM, l = 110 ± 20 nm (*n* = 100), is slightly smaller than the hydrodynamic radius viewed in DLS (210 ± 70 nm). Particles have length: thickness ratio of 1.96 ± 0.65, being relatively symmetrical. Pure MIL-88B-NH_2_ has been previously obtained but using DMF as reaction medium [27]. Taking into account DMF toxicity, the protocol proposed in this work for the first time with EtOH provides a safer alternative for industrial production of MIL-88B-NH_2_.

Another interesting parameter for improving STY is reaction time. As a consequence, kinetics studies were carried out using two temperatures (100 and 150 °C) to better analyze this variable (see Appendix A). Results revealed that, although reaction yield grew with incrementing reaction time, maximum STY was performed just after 5 min of reaction (16000 and 3300 kg·m^3^·d^−1^ at 100 and 150 °C, respectively).

DMF as reaction medium. As seen by the variety of phases obtained and characterized by PXRD, DMF is a complex reaction media. At concentrations below 0.2 M, iron trimers-based polymorphs (i.e., MIL-88B-NH_2_ and MIL-101-NH_2_) were favored, which seems to indicate that the formation of iron chains is only promoted in DMF at high concentrations. Regarding MIL-88B-NH_2_, even in mixture, was promoted with incrementing time within the 0.05–0.1 M range (see PXRD Appendix A). Formation of MIL-88B-NH_2_ over MIL-101-NH_2_ is not only modulated by the concentration, but also by the presence of strong acid as HCl. Reaction time was fixed for this test to be 30 min to maximize reaction yield (see Appendix A) using two different ligand:metal ratios. While the precursors ratio had no significant effect on the MOF formation, the presence of HCl affected the proportion of the resulting polymorphic phases, increasing the formation of MIL-88B-NH_2_ over MIL-101-NH_2_ (Appendix A). Furthermore, the presence of a strong acid produces MIL-88B-NH_2_ in DMF, while promoting MIL-53-NH_2_ in water, supporting the important role of the solvent on the nature of the obtained phases.

In addition, HCl modulates the crystal growth, procuring larger well-defined MIL-88B-NH_2_ crystals instead of MIL-101-NH_2_. (see Appendix A). Interestingly, MIL-88B-NH_2_ needles synthesized in DMF are 8 times longer than those formed in EtOH under similar conditions (without acid) and the length:thickness ratio is twofold times the EtOH ones (4.55 ± 1.42 *vs.* 1.96 ± 0.65, respectively). This indicates once again the important role of the reaction media on the formation of the different phases and its effect on the crystallinity and particle size. Of no less importance is the modulation effect provided by the presence of HCl in the mixture, as not only procured a pure MIL-88B-NH_2_ phase, but also modulates particle growth, having a length:thickness ratio of 3.11 ± 0.74, and half the length as in pristine DMF (410 ± 45 nm).

On the other hand, at high concentration (0.2 M), both MIL-53-NH_2_ and MIL-101-NH_2_ present a good crystallinity, favoring the MIL-53-NH_2_ thermodynamic phase with increasing time. Due to its flexible character, this phase could be better identified after drying the sample at 160 °C and removing the remaining EtOH and DMF from the flexible porosity (see Appendix A).

Finally, final yield increases with reaction time (see Appendix A). However, the mixture of phases limits further discussion.

Comparison of solvents: From these results, we can conclude that the reaction medium has the most relevant impact on the final phase attained due to several factors: solubility, de-protonation of the ligand (acid-base properties) and boiling T (and therefore final reaction pressure). As in our case, solubility of the H_2_BDC-NH_2_ ligand is really limited in water, even under reactions conditions (MW-assisted heat and pressure) and seems to decrease the final reaction yield. Nevertheless, H_2_O is a particularly interesting solvent as is a cheap, safe and green solvent with no security associated protocols rewarding its storage neither its manipulation, which is of particular interest for commercial protocols. What is more, water has the lower environmental, safety and health (ESH) impact, according to solvent selection guides of companies such as GSK or Pfizer [33]. In addition, under the studied conditions, water was the sole solvent that gave both pure MIL-101-NH_2_ and MIL-53-NH_2_ phases with highest crystallinity and relatively good STY (3850 and 10400 kg·m^−3^·d^−1^, respectively). While MIL-53-NH_2_ was obtained with water as reaction media for the first time, its polymorph, the MIL-68-NH_2_ still remains elusive under the studied variables.

EtOH is another interesting reaction medium, as is less harmful than other organic solvents and its recommended due to its low ESH. Compared to other organic solvents, with tendency to the crystallization of MIL-88B-NH_2_. In this work, it has been presented an outstanding STY of 16000 kg·m^−3^·d^−1^ at soft reaction conditions (T = 100 °C, t = 5 min), which exposes the beneficial use of MW-solvothermal assisted reaction for the industrial production of MIL-88B-NH_2_, which has demonstrated efficiency in applications such as photocatalysis, drug delivery and nanomotors [34,35,36,37]. In contrast, MIL-88B-NH_2_ particles with better crystallinity were produced in DMF, and should not be discarded as reaction medium only due to its known toxicity and cost compared with the previously mentioned solvents.

## 5. Conclusions

A thorough study has been carried out to analyze the influence of different reaction parameters on the MW assisted hydro/solvothermal synthesis of the FeCl_3_/H_2_BDC-NH_2_ isoreticular hybrid compounds. Three pure phases of iron(III) aminoterephthalate MOFs have been obtained, namely: MIL-53-NH_2,_ MIL-101-NH_2_ and MIL-88B-NH_2_. Characterization of the MOF phases by different solid-state techniques (PXRD, DLS, TEM) has enabled the identification of the reaction media as the main affecting variable of the MW assisted synthetic process. Importantly, the three different phases were obtained with water or either ethanol, both solvents with low ESH, relevant factor for MOF industrial production. Concentration of the MOF precursors and reaction temperature are other key parameters for phase selectivity. To the best of our knowledge, here is reported for the first time the MW assisted hydrothermal synthesis of under micron MIL-53(Fe)-NH_2_, opening the door for its industrial production.

## Figures and Tables

**Figure 1 materials-13-01469-f001:**
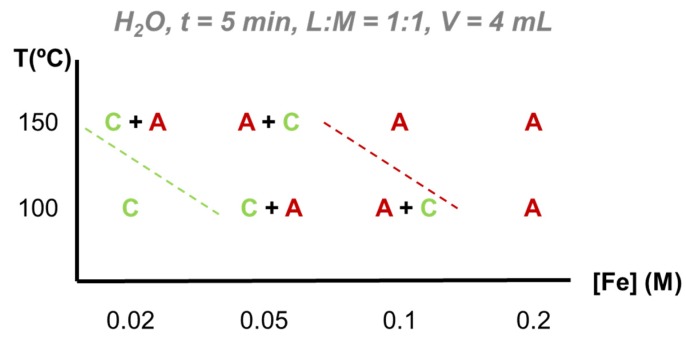
Crystallization diagram for the microwave (MW)-assisted hydrothermal synthesis of the system FeCl_3_/H_2_BDC-NH_2_ in water based on powder X-ray diffraction (PXRD) data. Legend: A = MIL-53-NH_2_, B = MIL-88B-NH_2_, C = MIL-101-NH_2_.

**Figure 2 materials-13-01469-f002:**
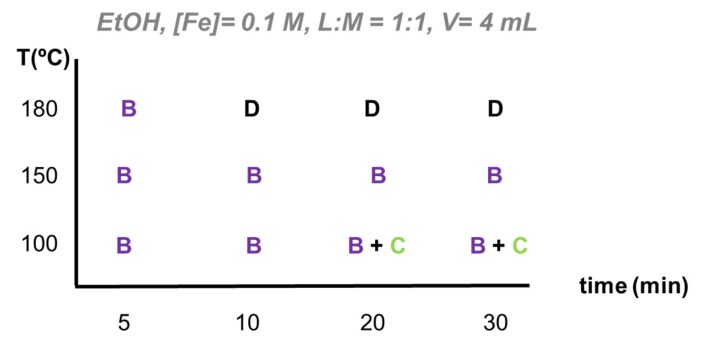
Crystallization diagram for the MW-assisted solvothermal synthesis of the system FeCl_3_/H_2_BDC-NH_2_ in EtOH based on PXRD data. Legend: A = to MIL-53-NH_2_, B = to MIL-88B-NH_2_, C = to MIL-101-NH_2_, D = Fe_2_O_3_.

**Figure 3 materials-13-01469-f003:**
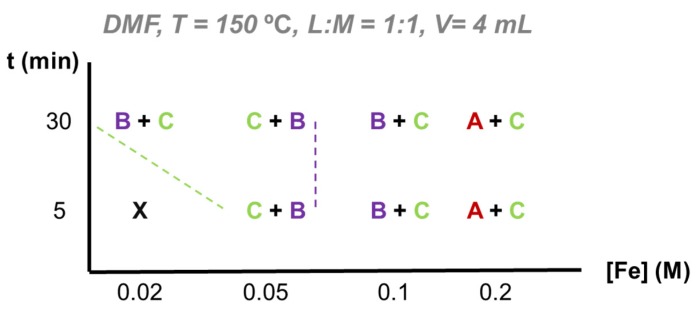
Crystallization diagram for the microwave-assisted solvothermal synthesis of the system FeCl_3_/H_2_BDC-NH_2_ in DMF based on PXRD data. Legend: A = to MIL-53-NH_2_, B = to MIL-88B-NH_2_, C = to MIL-101-NH_2_, X = amorphous.

**Figure 4 materials-13-01469-f004:**
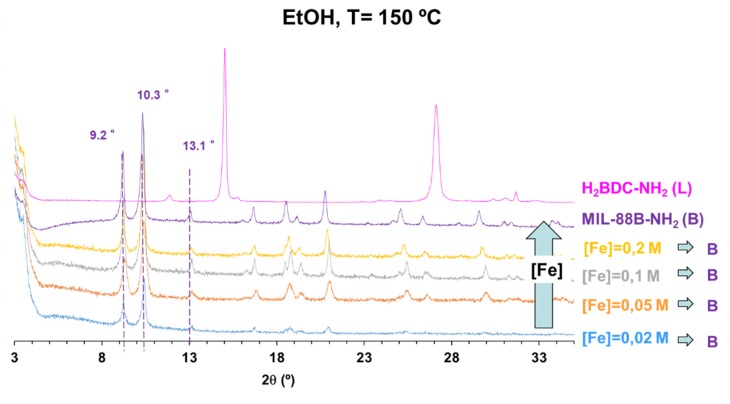
PXRD patterns for the MW investigation of the system FeCl_3_·6 H_2_O/H_2_BDC-NH_2_ in ethanol (V EtOH = 4 mL, T = 150 °C, t = 5 min) after activation with EtOH, compared to simulated MIL-88B-NH_2_ (purple) and H_2_BDC-NH_2_ (pink).

**Table 1 materials-13-01469-t001:** Mass, mol and molar ratios, and dispensed amounts for the MW investigation of the system FeCl_3_·6 H_2_O/H_2_BDC-NH_2_/HCl in water (V H_2_O = 4 mL, T = 150 °C, t = 5 min and ^(•)^t = 30 min). Legend: A = MIL-53-NH_2_, B = MIL-88B-NH_2_, C = MIL-101-NH_2_.

Sample Name	FeCl_3_ 6H_2_O (mg)	FeCl_3_ 6H_2_O (mmol)	H_2_BDC-NH_2_ (mg)	H_2_BDC-NH_2_ (mmol)	Ligand: Metal	[Fe] (M)	HCl 1 M (mL)	HCl: Fe	Result *
MW 2-01	21.6	0.08	14.48	0.08	1	0.02	0	0	**C + A**
MW 2-02	54	0.2	36.2	0.2	1	0.05	0	0	**A + C**
MW 2-03	108	0.4	72.4	0.4	1	0.1	0	0	**A**
MW 2-04 ^(•)^	108	0.4	72.4	0.4	1	0.1	0	0	**A**
MW 2-05	216	0.8	144.8	0.8	1	0.2	0	0	**A**
MW 2-06	54	0.2	36.2	0.2	1	0.05	0.1	0.5	**C + A**

* In the case of mixture, the first letter is the major phase.

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
