# Peer review of "Phase-Selective Microwave Assisted Synthesis of Iron(III) Aminoterephthalate MOFs"

_materials, 2020, doi:10.3390/ma13061469_

Round 1
Reviewer 1 Report
In this paper, the authors present microwave-assisted solvothermal (including hydrothermal) synthesis of several Fe-based MOFs, MIL-53, 88, and 101. The synthesis was controlled by varying solvent media, reaction temperature and time, Fe3+ concentration, and acid (HCl) concentration. They found that a specific condition can lead to a predominant product, although the MIL MOFs are normally obtained with the mixture. The experimental scheme and procedure seem to be highly systematic, and the paper is well organized and described. Only two things need to be corrected before publication.
- Sample codes (A, B, and C for MIL-53, 88, and 101) are used in most of the fiures in text. However, the codes are noted just in the supplementary information. It needs to be in the main text.
- microwave is a quite useful tool not only for synthesis but also for activation. Applications of the microwave for MOF needs in the introduction part. For the activation application: ACS Appl. Mater. Interfaces 2019, 11, 38, 35155-35161 (doi.org/10.1021/acsami.9b12201).
Author Response
We are very grateful for the pertinent comments and helpful suggestions proposed by the reviewer regarding our manuscript entitled " Phase-selective microwave assisted synthesis of iron(III) aminoterephthalate MOFs" by Ana Arenas-Vivo et al. (Reference: materials-746354), helping us to improve its quality.
As requested, please find below the reviewer comments in italics, and the corresponding responses, shown with blue lettering, including the modified text underlined in yellow.
____________________________________________
The experimental scheme and procedure seem to be highly systematic, and the paper is well organized and described.
We thank the reviewer for appreciating the results of the MW-assisted synthesis of Fe-MOFs. In the following, we address all her/his concerns.
- Sample codes (A, B, and C for MIL-53, 88, and 101) are used in most of the figures in text. However, the codes are noted just in the supplementary information. It needs to be in the main text.
R1. We apologize for omitting this information. All figures include now the following information: Legend: A= MIL-53-NH2, B= MIL-88B-NH2, C= MIL-101-NH2, X= amorphous, D= Fe2O3
- microwave is a quite useful tool not only for synthesis but also for activation. Applications of the microwave for MOF needs in the introduction part. For the activation application: ACS Appl. Mater. Interfaces 2019, 11, 38, 35155-35161 (doi.org/10.1021/acsami.9b12201).
R2. We thank Reviewer for the acknowledge of this interesting application of microwaves for MOF activation. This information has been included in the introduction as follows (line 72): “Even at its infancy, MW method has become a resourceful tool for the preparation of MOFs and for their activation / purification (removing pore filling species).[20,25]”
Reviewer 2 Report
The authors report a synthetic strategy together with MW-assisted hydro/solvothermal reactions to rapidly evaluate the influence of different reaction parameters (time, temperature, concentration, reaction media) on the formation of the benchmarked solids. Three pure phases of iron(III) aminoterephthalate MOFs have been obtained. Crystallinity analysis, as well as phase identification, has been carried out by PXRD, while particle size was investigated by DLS measurements and TEM. However, the manuscript needs to be improved before accepting it for publishing, and therefore I recommend major revision.
The following changes should be done:
01 Fig 1, 2, 3 - abbreviations are not clearly stated (A, B, C, L, M)
02 The discussion paragraph is very long and many pieces of information are repeating. The reader has the impression of many redundant data. Therefore the same data (particle size, HCl data could be grouped together.
03 Since the solvent media is the key component and greatly impacts the final phase, more solvents should be tested (e.g. set of alcohols, methanol, ethanol, isopropyl alcohol), so that the proposed research has relevant feedback. Also, I suggest testing polar protic and polar aprotic solvents to make a comparison in that regard.
Author Response
We are very grateful for the pertinent comments and helpful suggestions proposed by the reviewer regarding our manuscript entitled " Phase-selective microwave assisted synthesis of iron(III) aminoterephthalate MOFs" by Ana Arenas-Vivo et al. (Reference: materials-746354), helping us to improve its quality.
As requested, please find below the reviewer comments in italics, and the corresponding responses, shown with blue lettering, including the modified text underlined in yellow.
01 Fig 1, 2, 3 - abbreviations are not clearly stated (A, B, C, L, M)
R1. We apologize for omitting this information. All figures include now the following information: Legend: A= MIL-53-NH2, B= MIL-88B-NH2, C= MIL-101-NH2, X= amorphous, D= Fe2O3
02 The discussion paragraph is very long and many pieces of information are repeating. The reader has the impression of many redundant data. Therefore the same data (particle size, HCl data could be grouped together.
R2. We agree that the discussion section is long as it includes a detailed analysis of the 46 individual reactions carried out in this study. Results have been organized for clarity regarding the different solvent used, as it was found during this study that the reaction medium has the most relevant impact on the final phase attained due to several factors: solubility, de-protonation of the ligand (acid-base properties) and boiling temperature (and therefore final reaction pressure). Therefore, powder x-ray diffraction (PXRD) patterns, dynamic light scattering (DLS) measurement and particle size analysis with transmission electron microscopy (TEM) and quantification of the space time yield (STY), are analysed accordingly inside each section. We consider that this discussion should not be omitted because, as stated in the conclusion: “Characterization of the MOF phases by different solid-state techniques (PXRD, DLS, TEM) has enabled the identification of the reaction media as the main affecting variable of the MW assisted synthetic process”. In addition, all resumed details have been now resumed in the supporting information in Tables S9-S17
03 Since the solvent media is the key component and greatly impacts the final phase, more solvents should be tested (e.g. set of alcohols, methanol, ethanol, isopropyl alcohol), so that the proposed research has relevant feedback. Also, I suggest testing polar protic and polar aprotic solvents to make a comparison in that regard.
R3. We totally agree with the reviewer that the solvent used in microwave assisted reactions is a key parameter in the determination of the final phase obtained. In accordance, it is also one of the main conclusions of our work. Nevertheless, we have chosen those three solvents among all as both water and ethanol are considered at industrial level to have low environmental, safety and health impact as stated by Chemical companies selection guides (GSK, Pfizer). On the other hand, DMF was selected as it is a common solvent in MOF synthesis. In addition, from literature, these solvents have been widely proposed in the crystallization of iron carboxylates, in particular, in the synthesis of MIL-53, MIL-88B and MIL-101 structures. These selection criteria and the importance of water and ethanol as green solvent in order to MOF synthesis scale-up has been included in the main text as follows:
Line 204 “Water and ethanol have been selected as good examples of green chemistry reaction solvents. DMF have been selective as a typical reaction media traditionally used in MOF solvothermal synthesis.”
Line 321 “What is more, water has the lower environmental, safety and health (ESH) impact, according to solvent selection guides of companies such as GSK or Pfizer.[32]”
Line 327 “EtOH is another interesting reaction medium, as is less harmful than other organic solvents and its recommended due to its low ESH.”
Line 342 “Importantly, the three different phases were obtained with water or either ethanol, both solvents with low ESH, important factor for MOF synthesis industrialization.”
We recognize the interest of further studies with both protic and aprotic solvents, even for obtaining MOF phases potentially different to MIL-53, -88B and -101, but due to the extension of the present work, they need to be postponed to further research works.
Reviewer 3 Report
The paper is well presented and reports interesting results. For clarity, it must be completed with some results now placed as supplementary information.
- “…to achieve their commercial production…” The authors should explain how a MW synthesis is closer to some kind of up-scaling that a conventional batch reactor. Perhaps the authors can do mention to some existing medium-large scale MW system.
- “Even at its infancy, MW method has become a resourceful tool for the preparation of MOFs.” Some general wording about the general advantages of MW as compared to others synthesis (solvothermal, ultradounds, solvent-free, etc) could help to clarify the scope of the paper.
- On the contrary, the use of green solvents such ethanol or water brings lab and industrial practice closer, and it should be highlighted.
- When calculating the MOF yield with eq 1 the mass of MOF should be considered in dry basis, excluding solvent and possible ligand trapped. For that TGA information should be applied.
- 1-3 captions should indicate what the capital letters are.
- In fact, for clarity some of the material from the SI file should be placed in the main paper: Table S1, a simplified table of experiments from Tables S2 to S8 and some key XRD patterns. Otherwise the paper is hard to follow.
Author Response
We are very grateful for the pertinent comments and helpful suggestions proposed by the reviewer regarding our manuscript entitled " Phase-selective microwave assisted synthesis of iron(III) aminoterephthalate MOFs" by Ana Arenas-Vivo et al. (Reference: materials-746354), helping us to improve its quality.
As requested, please find below the reviewer comments in italics, and the corresponding responses, shown with blue lettering, including the modified text underlined in yellow.
The paper is well presented and reports interesting results. For clarity, it must be completed with some results now placed as supplementary information.
We thank the reviewer for appreciating the results of the MW-assisted synthesis of Fe-MOFs. In the following, we address all her/his concerns.
- “…to achieve their commercial production…” The authors should explain how a MW synthesis is closer to some kind of up-scaling that a conventional batch reactor. Perhaps the authors can do mention to some existing medium-large scale MW system.
R1. We thank reviewer for this appreciation. Further information has been included in the introduction section of the article (starting in line 67) as follows: The conjunction of this assets dramatically increases production due to the homogeneous energy input, compared to a traditional batch reactor, which can even be enhanced by reaction stirring.[21] In addition, industrial production in the order of T·year-1 can be achieved under MW-assisted continuous flow synthesis.[22] In consonance, this technology is efficiently exploited nowadays in the production of several organic chemicals such as drugs or polymers.[23,24]
- “Even at its infancy, MW method has become a resourceful tool for the preparation of MOFs.” Some general wording about the general advantages of MW as compared to others synthesis (solvothermal, ultradounds, solvent-free, etc) could help to clarify the scope of the paper.
R2. Further stress has been put into the description of MW synthesis over other traditional synthetic methods in the previous paragraph (starting in line 63), as follows: With this is mind, microwave synthesis (MW) of MOFs has been proposed as an alternative to conventional hydro- or solvothermal reactions due to several advantages: i) energy efficiency, ii) fast crystallization (increment in number of reaction sites), iii) phase selectivity, iv) high yields, v) variety of morphologies, vi) particle size control, vii) lower temperatures and reaction times.[20]
- On the contrary, the use of green solvents such ethanol or water brings lab and industrial practice closer, and it should be highlighted.
R3. We agree with reviewer that it is important to emphasize the results obtained with those green solvents. This part has been stressed in the main text and in the conclusion part as follows: Line 321“What is more, water has the lower environmental, safety and health (ESH) impact, according to solvent selection guides of companies such as GSK or Pfizer.[32]”; line 327 ” EtOH is another interesting reaction medium, as is less harmful than other organic solvents and it is recommended due to its low ESH”; line 204 “Water and ethanol have been selected as good examples of green chemistry reaction solvents. DMF has been selected as a typical reaction medium traditionally used in MOF solvothermal synthesis”. And line 341 . “Importantly, the three different phases were obtained with water or either ethanol, both solvents with low ESH, important factor for MOF industrial production”.
- When calculating the MOF yield with eq 1 the mass of MOF should be considered in dry basis, excluding solvent and possible ligand trapped. For that TGA information should be applied.
R4. We thank reviewer for pointing out this detail. We want to clarify that the parameter was determined after drying the MOF sample at 100 °C in a furnace. This relevant information has been included within the main text that now reads (line 125): “where mFeBDC-NH2 is the experimental mass of the Fe-BDC-NH2 MOF obtained (determined after drying at 100 °C)”
- 1-3 captions should indicate what the capital letters are.
R5. We apologize for omitting this information. All figures include now the following information: Legend: A= MIL-53-NH2, B= MIL-88B-NH2, C= MIL-101-NH2, X= amorphous, D= Fe2O3
- In fact, for clarity some of the material from the SI file should be placed in the main paper: Table S1, a simplified table of experiments from Tables S2 to S8 and some key XRD patterns. Otherwise the paper is hard to follow.
R6. For helping the reader with the better understanding of the results obtained in this work, some pieces of information have been moved to the main text. Regarding the tables with the reaction conditions, just one table (Table 1) has been included in the main text as example to keep the reading fluency. The same criteria has been followed with the PXRD patterns (Figure 4).
Round 2
Reviewer 2 Report
The authors corrected the things that were asked and they have improved the manuscript. For that reason, I suggest acceptance of the manuscript.